# The Effects of Sickle Cell Disease on the Quality of Life: A Focus on the Untold Experiences of Parents in Tanzania

**DOI:** 10.3390/ijerph19116871

**Published:** 2022-06-04

**Authors:** Manase Kilonzi, Dorkasi L. Mwakawanga, Fatuma Felix Felician, Hamu J. Mlyuka, Lulu Chirande, David T. Myemba, Godfrey Sambayi, Ritah F. Mutagonda, Wigilya P. Mikomangwa, Joyce Ndunguru, Agnes Jonathan, Paschal Ruggajo, Irene Kida Minja, Emmanuel Balandya, Julie Makani, Nathanael Sirili

**Affiliations:** 1School of Pharmacy, The Muhimbili University of Health and Allied Sciences, Dar es Salaam P.O. Box 65013, Tanzania; fatmafelix@yahoo.com (F.F.F.); hmlyuka2011@gmail.com (H.J.M.); dmyemba09@gmail.com (D.T.M.); mzalendohalisi08@gmail.com (G.S.); rittdavisrida@yahoo.com (R.F.M.); wpad.miko@gmail.com (W.P.M.); 2School of Nursing, The Muhimbili University of Health and Allied Sciences, Dar es Salaam P.O. Box 65013, Tanzania; dorkasmwakawanga@gmail.com; 3School of Medicine, The Muhimbili University of Health and Allied Sciences, Dar es Salaam P.O. Box 65001, Tanzania; chirandelulu@yahoo.com (L.C.); jnduguru@blood.ac.tz (J.N.); ajonathan@blood.ac.tz (A.J.); prugajo@gmail.com (P.R.); ebalandya@yahoo.com (E.B.); jmakani@blood.ac.tz (J.M.); 4School of Dentistry, The Muhimbili University of Health and Allied Sciences, Dar es Salaam P.O. Box 65013, Tanzania; ikminja@gmail.com; 5Department of Development Studies, School of Public Health and Social Sciences, Muhimbili University of Health and Allied Sciences, Dar es Salaam P.O. Box 65013, Tanzania; drnsirili@gmail.com

**Keywords:** SCD, children, parents, impact, quality of life, Tanzania

## Abstract

Tanzania is among the top five countries with a high burden of sickle cell disease (SCD) in the world. Even though the effects of SCD on quality of life have been documented in other countries including Nigeria and the United States of America, few are known from Tanzania. Therefore, this study focused on evaluating the effects of SCD on the quality of life among children living with SCD and their parents. The study employed a qualitative approach to interview purposively selected parents of children who have lived with SCD and have used hydroxyurea (HU) for more than 3 years. The in-depth interviews were conducted with 11 parents of children with SCD at the Muhimbili University of Health and Allied Sciences (MUHAS) in Dar-es-salaam, Tanzania. A semi-structured interview guide was used. Interviews were audio-recorded, transcribed, and thematically analyzed. Three themes were generated including psycho-social effects: family conflicts and divorce, limited access to education, stress and fear; financial effects: Employment limitation, reduced efficiency and productivity, loss of job and lack of self-keeping expenses; and physical effects: physical disability and dependence, and burden of the frequent crisis. Children living with SCD and their parents suffer psycho-social, financial, and physical impacts of the disease. Appropriate interventions should be introduced to minimize the observed effects as ways of improving the quality of life of the individuals living with SCD and their caregivers.

## 1. Introduction

Sickle cell disease (SCD) is a devastating hematologic disorder caused by a single nucleotide mutation that leads to formation of abnormal hemoglobin S which has the tendency to polymerize and cause sickling of erythrocytes [1]. It is estimated that about 300,000 babies are born with SCD globally, with the majority (75%) being in Africa [2]. Tanzania, which ranks fifth worldwide in terms of SCD birth prevalence, has 11,000 children born with SCD every year [3]. Unfortunately, in the absence of proper care, up to 90% of these children could die before reaching 18 years of age. While hemoglobin disorders represented 3.4% of the under-five mortality worldwide, in Tanzania, SCD alone contributes to 6.4% of the total under-five mortality [4].

SCD impacts the well-being of individuals and their quality of life in general. Studies have narrated how SCD is a burden to patients’ physical and mental health, social life, school, and work. For instance, affected individuals can experience mental health disorders such as anxiety of death, denial about their disease, and depression. Moreover, problems of self-esteem and body image concerns due to poor growth and abnormalities are common to individuals with SCD and as a result, they have limited interactions with their healthy peers [5]. Furthermore, low or lack of self-esteem has interfered with the ability of individuals with SCD to form significant relationships. Research further describes frequent hospitalization interfering with school attendance, resulting in poor school performance and educational status of the affected individuals [6,7].

Despite having a significant number of individuals with SCD in Sub-Saharan Africa (SSA), only a few countries in the region have documented the effects of SCD on the quality of life among children living with SCD and their parents [8,9]. The information is crucial in facilitating the design of appropriate interventions to improve the quality of life of individuals with SCD. Therefore, this study aimed to evaluate the effects of SCD on the quality of life among children living with SCD and their parents in Tanzania.

## 2. Methods

### 2.1. Study Design and Setting

This research employed the explorative qualitative approach that explored the effects of SCD on the quality of life among children living with SCD and their parents in Tanzania. The Muhimbili National Hospital (MNH) had established an SCD clinic in 1980 that served patients within and outside the business capital of Dar-es-salaam [4]. In 2004, the Muhimbili University of Health and Allied Sciences (MUHAS) integrated the MNH SCD clinic with research [4]. Regardless of the efforts, the majority of individuals with SCD in Tanzania still have to travel long distances to attend SCD clinics.

This study was conducted between November 2020 and January 2021 to evaluate the effect of SCD on the quality of life among children living with SCD and their parents. Interviews were conducted at MUHAS, which is the premier medical university in Tanzania. For the purpose of this study, parents were asked to visit the university for an in-depth interview.

### 2.2. Study Participants and Recruitment

A total of 11 parents of children living with SCD participated in this study (8 females and 3 males) with a median age of 42 years ranging from 24 to 52 years. Participants were purposively recruited from the electronic database of the Sickle Pan-African Research Consortium (SPARCO)-Tanzania database at MUHAS (patients’ registry). The study was part of the project that assessed factors influencing the utilization of HU in Tanzania [10]. Therefore, parents with SCD children who never used HU were excluded. Parents of children with SCD who were on HU for at least 3 years with or without health insurance, and those who had used HU for at least 3 years but had stopped, were recruited. The parents were selected to gain insight on the effects of SCD on their quality of life and that of their children who are living with SCD. Before commencement of data collection, the study obtained institutional permission, ethics committee permission, and informed consent from the participants. All 11 approached parents consented to participate, none denied.

### 2.3. Data Collection Procedures

We developed a semi-structured interview guide based on the literature review and investigators’ experiences on the subject. Among the issues asked during the interview, some included: general understanding of the parents with regard to SCD (some of the probe questions are meaning, causes, transmission, treatment and prevention of SCD). Another question asked about how SCD has impacted his/her life and the children, with probes directed towards; economic impacts (work, income, expenditure), social impacts (education, family life, sports, recreation) and psychological impacts (perception about life, feelings, stress). We collected data using a semi-structured interview guide in the Kiswahili language which is a native language for both researchers and participants. The interviews were conducted by two researchers who are also authors of this article (MK and HJM). Further information regarding the approval, data collection procedures, and how the in-depth interviews were conducted have been explained in our previous published paper [10].

### 2.4. Data Analysis

A thematic approach was adopted to analyze the collected information. Audio-recorded interviews were first transcribed verbatim manually. Before the coding process, the full transcripts and field notes were read and re-read by all authors to become familiar with the data and the context. Deductive–inductive, team-based coding was implemented for codebook development and data analysis. Coding process was conducted by two groups supervised by the first and fourth authors. To avoid discrepancy, the first two transcripts were coded by each group separately, and then they were compared for agreement on the final codes and coding. The generated codes were grouped into the respective pre-determined codes through comparisons. Then, the frequency of appearance of the related codes generated a sub-theme and finally, themes were developed following the contextualization and conceptualization process.

### 2.5. Rigor and Trustworthiness

To ensure the trustworthiness of this study, we used four criteria of credibility, dependability, transferability and conformability. Credibility of the findings was ensured through the use of purposeful sampling approach to recruit study participants. The use of Kiswahili language also enhanced the credibility of the findings given the fact that Kiswahili is the national language and is spoken comfortably by the majority of study participants which was important to limit the risk of misinformation. Dependability was ensured by doing initial analysis by all authors and thereafter agreeing on the codes. The use of a thematic analysis approach and the involvement of different investigators with different expertise during data analysis managed the team to generate appropriate code and finally sub-themes and themes. The description of the study design and setting, the purposive sampling approach used, the data collection method that was used, the description of the analysis process, and the use of participants’ accounts allowed for us to determine the study’s transferability to other families, communities, countries, and cultures with a high burden of SCD.

## 3. Results

We interviewed 11 parents of children with SCD, of which 8 were females and 3 were males. The median age of the parents was 42 years with a range of (24–52) years. Except for one parent, the rest were married, and the majority had attained primary or secondary education and were self-employed. Two parents had children who were not covered by health insurance. From the analysis of interviews, three themes of the impacts of SCD on the quality of life of the children with SCD and the parents/guardians in Tanzania were revealed. These were: psycho-social-related impacts, financial-related impacts, and physical impacts (Figure 1).

### 3.1. Psycho-Social Related Impacts

The quality of life on the psychological and social part of the children and their parents was reported to be affected due to many factors. The interviews revealed family conflicts and divorce, limited social interactions, limited access to education, and fear and stress as major impacts affecting the psychosocial life of the children with sickle cell and parents.

(a)Family Conflicts and Divorce

Participants reported going through difficult situations due to having a child with SCD in the family. They expressed that the presence of an individual with SCD in the family has introduced misunderstanding between parents resulting in divorce, abandonment by a male parent, and even the introduction of extramarital relationships as an escape not to be close to the woman having a child with SCD. On the other hand, parents reported having a limitation in establishing relationships due to worries and fear of having a child with SCD. One participant said:


*“… It is discouraging, like how can you have a baby? Sometimes when you are in a relationship, you’re afraid of becoming pregnant due to chances of having a child with SCD which most of the time leads to breakup”.*
Parent 7

Another participant stated that children have been limited in participating in some of the school social activities such as sports due to frequent hospital admissions and pain which has also introduced psychological disturbances among children with SCD. The parent said;


*“… When he got a stroke, he couldn’t walk. He has been recovering but sometimes when he goes alone to school and meets his friends or just people, he would face some stigma. He would come back complaining that people have been laughing at how he walks, teasing him…a lot of things happen but I encourage him. I tell him to recognize his progress and be grateful to God”.*
Parent 11

(b)Limited access to education

Parents reported that SCD has negatively impacted children’s education paths due to the inability to attend school, delays to start studies, and being limited to stay in boarding schools. The absenteeism was explained to be due to repeated hospital admission for different types of crises. Parents believe that completing studies is very important as an assurance of quality life to children. This has caused psychological disturbances and has been reported to be one of the major stresses to children and their parents as well. One participant said;


*“… Regarding education, perhaps I should say that I find it tough to send my child to a boarding school, because am worried that she might fall sick the minute I leave her there”.*
Parent 1

Moreover, participants expressed those children with SCD have been facing mistreatment and discrimination at school. Other children have been labeled as SCD children for identification which has negatively affected their emotions and social life. Some parents reported that children drop out due to poor performance at school as well as due to the feeling that education is becoming a burden when the child’s condition is not encouraging. One participant said;


*“… when he got a stroke, he couldn’t walk. He has been recovering but sometimes when he goes alone to school meets his friends or just people, he would face some stigma. He would come back complaining that people have been laughing at how he walks, teasing him…a lot of things happen but I encourage him I tell him to recognize his progress and be grateful to God”.*
Parent 11

(c)Fear and Stress

Participants reported that having a child with SCD has introduced stress and fear into their life. Some participants reported having fear of marriage, fear of having more children, and even fear of death of their children. These have caused mental disturbances in families that have reduced their confidence in life. They further reported to have fear of financial sustainability that they may not be able to meet the demand of taking care of the child with SCD.


*“… Psychologically am affected because I think a lot on how to take care of my sick child because her health insurance is expired, my business is not doing well, she needs to go to school and all of these need money”.*
Parent 4

### 3.2. Financial Related Impacts

Parents reported the financial constraints due to care for children with SCD due to limitations to employment, loss of jobs, reduced productivity, and efficiency that led to a lack of self-keeping expenses.

(a)Employment limitation

Many participants complained that they had very little opportunities to apply for jobs because of the little time they can spend at work. They explained that most of their time is spent caring for a child with SCD and they have limited chances for employment in jobs which have demanding goals and are time consuming. Further, they stated that even a child with SCD grows up with a very narrow chance to get employment because of being unstable physically.


*“… This affects business since I can’t work when my child is sick. I have thought of finding a job but then when I think of my child’s condition, I feel like it won’t work. I would end up being fired due to regular absenteeism”.*
Parent 4

(b)Loss of Job

The financial burden for most of the participants was due to loss of job whereby those who had been working previously had lost their job. This was because parents dedicated most of their time to tend to their sick children and had several absences from work which was intolerable to the majority of their employers. This brought challenges to the majority of parents of being unable to afford the expenses for frequent hospitalizations and medical bills, which all require money.


*“… sometimes when am home I will just receive a call from school that my child is sick and I should go pick him up. Therefore, I will have to leave work and send my child to the hospital where we will spend a week or two”.*
Parent 4

(c)Reduced Efficiency and Productivity

Having a child with SCD reduced the productivity of most of the parents. They reported having less time to engage in different income-generating activities such as farming and entrepreneurship that would help in health expenditures for the child and housekeeping expenses. The less involvement in work had resulted in a substantial reduction of productivity. Participants expressed that the latter had led to an economic drop-down for families with an individual with SCD.


*“… Talking of economic capability, am financially unstable. It was tough finding a job because I couldn’t find time to do so. I had to self-employ”.*
Parent 3

(d)Lack of self-keeping expenses

Participants reported having a greater amount of family income spent on health care needs for the child with SCD that negatively affected the family in terms of the ability to meet the daily living expenses. They stated that having lost their jobs, the reduced work productivity and employability had also led their families to inability to cover the household expenses. One participant said;


*“… Financially am failing because during pain episodes or when he is sick, I can’t go to work. The good thing is I am self-employed. I think if I was employed by someone, I would have been fired due to regular absenteeism from work”.*
Parent 11

### 3.3. Physical Related Impacts

As reported in this study, SCD is responsible for several health challenges and medical complications that have affected the quality of life of children and families. From parents, we found that children with SCD were burdened by the physical disability, dependence, and frequent crises.

(a)Physical disability and dependence

Some participants reported having children who have a physical disability due to stroke. They expressed that SCD has negatively affected their children’s ability to perform physical activities which in turn has increased the burden of dependence on their caregivers and/or parents. Therefore, participants felt that having a disabled child created a burden to the family and felt uncomfortable when they are visited by people at home.


*“… He suffered stroke severally. I remember he encountered stroke at three different times. During the first time, he paralyzed both extremities. The second time he paralyzed his mouth and arms”.*
Parent 6

(b)The burden of the frequent crisis

The medical effect of SCD, that results in frequent vaso-occlusive crisis, manifested as an absolute challenge among the majority of parents. Parents stated that they were not well-prepared psychologically that their children will be having frequent crises for their entire life. Therefore, it has been tough for them as parents to witness the pain and sufferings their children undergo.


*“You may find him with swollen fingers, and sometimes swollen thigh. For those who are aware of the symptoms, they don’t panic… sometimes he complains of leg pain and difficulty in walking”.*
Parent 11

## 4. Discussion

We aimed to explore the impacts of SCD on the quality of life of parents and children with SCD in Tanzania. Our findings have unveiled three themes, which are the psycho-social, financial, and physical impacts of SCD.

An individual with SCD suffers frequent vaso-occlusive crises which necessitates frequent hospital visits and hospitalization, which have also been reported in this study. The parents reported that SCD poses a physical impact on their children. Parents emphasized the children are at risk of stroke, and if this happens the children become dependent. The findings are supported by previous studies in which among the issues that affected the health-related quality of life of the individuals with SCD included physical disability due to stroke which limits the children from playing and participating in household activities.

In Nigeria, it was reported that children with SCD are missing school and have poor school performance due to the physical impacts of SCD [11,12]. Additionally, in some schools, special labels are placed on the clothes of pupils/students with SCD for easy identification. This approach has added a problem of stigmatization which limits pupils/students’ school attendance and their performance. Furthermore, pupils/students with SCD face rejection from other students during social activities and gatherings. The findings are consistent with what has been reported in a previous study conducted in developing countries whereby students with SCD reported being excluded from playing and faced bullying from other students because of their yellowish eyes [9].

Parents, particularly mothers, reported facing family conflicts and divorce because of having a child with SCD. Some of the participants explained that some of the spouses believe that having a child with SCD is linked with curses, witchcraft, and bad luck. Poor knowledge, perception, and a negative attitude towards SCD have been associated with several challenges among children with SCD and their caregivers [13]. Studies demonstrated that individuals with SCD and their caregivers are facing denunciation from both family and community [14,15]. A large population of people living in developing countries has a low level of education which limits their understanding of many diseases and ends up connecting hereditary diseases with curse and witchcraft [16,17]. Due to poverty, some people in developing countries either have no formal education or only have a primary education [18,19]. This necessitates the need for awareness campaigns to educate the community regarding hereditary diseases such as SCD.

Our study revealed that both parents and children with SCD are living with stress and fear including fear of marriage, fear of having children, and fear of death. The findings are in line with what was reported in similar studies in which anxiety (21–33%) and depression (7–36%) were the common reported psychological impacts facing individuals living with SCD [20]. In addition, studies have reported that children with SCD are worried about talking to their friends and teachers regarding their problem and are in fear of interacting with others because of their yellowish eyes (jaundice) that may be identified [9]. If the situation is left unattended, individuals with SCD are at risk of engaging in drug abuse and committing suicide due to feelings of isolation and depression.

Individuals with SCD have to deal with frequent hospital visits, hospitalization, and comorbidities such as bacterial infections, malaria, and anemia [21,22]. Parents revealed that they are facing economic burdens resulting from healthcare demands for their children with SCD. Parents stressed that they often run out of money because they have limited time for productivity and it is difficult for them to be employed. The findings correlate with previous studies whereby caregivers and individuals with SCD reported facing difficulty in performing for their business/job, a loss of productive time, and facing bankruptcy because of taking care of the hospital bills [23,24]. It has been reported in Nigeria that some of the parents ended up selling their assets and belongings and received loans to be able to clear hospital bills [8].

Nevertheless, to fight against health expenditures costs, WHO emphasize the use of healthcare insurance [25]. The idea of universal healthcare coverage is a working promise in developed countries but the opposite is true in developing countries [8,26]. In low–middle income countries, the majority of the people rely on out of pocket approach in paying hospital bills rather than the pre-payment approach [8]. The approach renders them in financial problems which sometimes forces them to sell their assets, belongings, and take out unnecessary loans [27,28,29]. Stakeholder engagement is needed to ensure that parents with SCD children, and the population in general, understand the need for having health insurance, the benefits of health insurance, and where to access health insurance services.

We acknowledge that the study missed some non-verbal communication due to the online interview with some parents, Additionally, the study had a small sample size of 11 and the interview was offered only in Swahili, thereby selecting in a biased manner those who were able to participate.

## 5. Conclusions

Children living with SCD and their parents suffer psycho-social-, financial-, and physical-related impacts of the hereditary disease. Counselling and psychosocial support should be routinely provided to children with SCD and their families. Forming patient/parent support groups and linking families to civil societies and NGOs with interest on SCD is another way of supporting these families.

## Figures and Tables

**Figure 1 ijerph-19-06871-f001:**
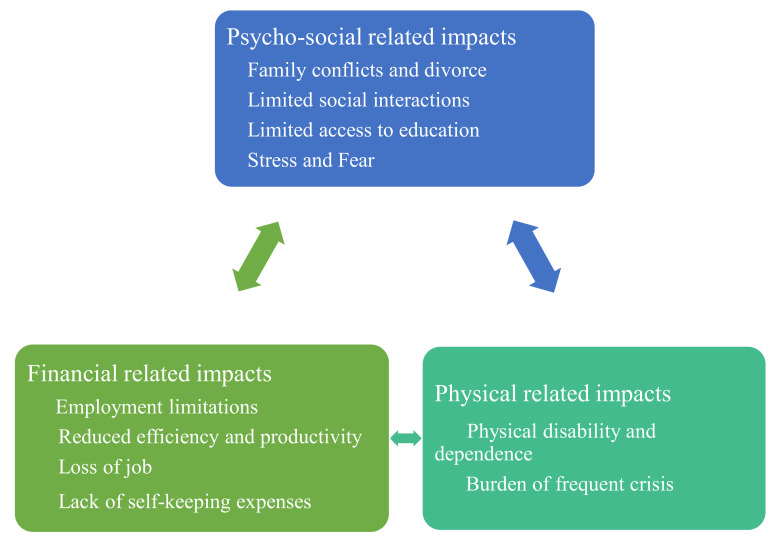
Summary of findings on the effect of SCD on the quality of life among children living with SCD and their parents.

## Data Availability

Data are not available publicly because they contain sensitive interview information and participants did not consent for their interviews to be shared publicly. The data are available from The Directorate of Research and Publication Muhimbili University of Health and Allied Sciences (contact via: drp@muhas.ac.tz Tel.: +255-21503026) for researchers who have criteria to access confidential information.

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
