# Peer review of "The Effects of Sickle Cell Disease on the Quality of Life: A Focus on the Untold Experiences of Parents in Tanzania"

_ijerph, 2022, doi:10.3390/ijerph19116871_

Round 1

Reviewer 1 Report

Thank you for the opportunity to review the manuscript entitled, "The effects of Sickle Cell Disease on the Quality of Life: A Focus on the Untold Experiences of Parents in Tanzania". This is a qualitative study of 11 parents about the effects of SCD on their quality of life.  Although, the findings are not novel, it is still critical to confirm them in various cultures around the globe in order to improve care within each cultural context.  As such, this is an important study for children living with SCD in Tanzania.  Before it can be accepted, several improvements can be made:

Intro:

  • Since you include interview parents who children took or are taking HU for at least 3 years, please explain why this is important in your intro.
  • Your aim currently is "to evaluate the effects of SC on the qulity of life among children living with SCD and their parents in Tanzania"... which could be done by interviewing any parent with SCD, so why limit to only those who took or are taking SCD.

Methods:

  • How did you identify the patients who took or are taking HU for three years? Did you cross reference your SCD patients with some pharmacy data?
  • Did you approach every eligible patient as they came into clinic between 11/20 and 1/21 or how did you identify which parents were approached to participate in the study?
  • Very importantly, did you record why parents refused to consent or at least who they were to study if they biased your results?  If you didn't track this, do you know how many you approached and how many declined?  If you know that, please report it in the results section.
  • "The interviews were conducted by two researchers which are also authors" should be "who are also authors"
  • Since you give quotes in the results section, who translated and when were the interviews translated to English? (i.e.: before or after coding).

Results

  • Results were relatively straightforward and easy to read. I appreciated the quotes as they give you a glimpse into the lives of these parents.

Discussion

  • "Special labels are placed in the cloth of pupils" should be "special labels are placed on the clothes of pupils" in American English. Not sure if "in the cloth" is an accepted phrase in English from other countries.
  • "Participants explained that some of the spouses believe that having a child with SCD is linked with curse" is a very interesting and scary statement.  Did none of the parents who participated in the interviews believe this? Why not and why only the spouses?  Did the clinic educate the one parent in clinic about this and the other parent never heard it? If so, then how can both parents be educated about this stigmatizing myth?  How many spouses of the 11 thought this?   
  • "they often run out of pockets" is not a phrase used in American language so I don't understand it.  Do you mean that they often run out of money.  If so, I recommend replacing pocket with money
  • Can paragraphs 4 and 5 in the summary be combined somehow.  They seem redundant about the economic burden
  • The sentence that starts with "The use off HU by individuals with SCD has been associated with several benefits including..." should have citations
  • How you cite your references vary. For example, you have "[26][27][28]" in one place and "[29]-[31]" in another
  • "These include: misconceptions of parents on SCD". I think this should read "These include: misconceptions of parents of children with SCD", yes?
  • "off pocket" is not a term in US English.  We would use "Out of pocket" to mean that insurance does not pay for it and we must take cash out of our pockets to pay for it.
  • Under Rigor and Trustworthiness, please add a sentence about:
    • Generalizability of this data: to which families, communities, countries, and/or cultures?
    • Bias: specifically, who declined participating and therefore are missing from these data completely?  In general, parents who are more educated and have the time to participate will consent.  I very much worry who refused to participate and how this could significantly skew the data, especially given only n=11.

Author Response

Thank you for the comments and suggestions. Through your comments, we have improved the draft.

Author Response

Thank you for the comments and suggestions. We have addressed all comments

Reviewer 3 Report

The authors have described The effects of Sickle Cell Disease on the Quality of Life: A Focus on the Untold Experiences of Parents in Tanzania clearly and concisely. 

  1. The authors should report if the study was IRB approved and if yes, the what institution?
  2. The authors should list the small sample size as limitation. 
  3. Did authors surveyed any patients on chronic transfusion?
  4. The authors should state the future directions, how this report can help to improve quality of life and what measures should be taken?
  5. The authors should list the name of software they used for translation and transcription?
  6. Did the authors detect the interreviewer variability?

Author Response

Thank you for the nice comments and suggestions. We have addressed all the issues raised in the attached word document.

Round 2

Reviewer 1 Report

Thank you for addressing our suggestions.  I believe this can be accepted after some minor edits.

I suspect the edits in red were completed hastily and there appears to be fragment sentences. Please read the whole paper thoroughly. For example "Credibility of the findings was ensured through the use of purposeful sampling approach was used to recruit study participants and the use of Kiswahili language to conduct (native language for researchers and participants)" looks like what is printed but is both/either a run-on sentence or a long sentence fragment.

Limitations paragraph: This should be moved to the discussion section.  Small sample size of 11 and that the interview was offered only in Swahili, thereby selecting and biasing who was approached to participate, needs to be added.  Furthermore, pharmacists can be sympathetic and empathetic so I would delete that part. This paragraph is awkwardly worded so should be proofread as suggested above. 

I do not need to re-review as long as technical writers and editors agree that grammar is acceptable.

Author Response

Thank you for the nice suggestions. We have addressed all the comments/suggestions in the main document using track changes.
